# Color-Flu Fluorescent Reporter Influenza A Viruses Allow for In Vivo Studies of Innate Immune Function in Zebrafish

**DOI:** 10.3390/v16010155

**Published:** 2024-01-20

**Authors:** Brandy-Lee Soos, Alec Ballinger, Mykayla Weinstein, Haley Foreman, Julianna Grampone, Samuel Weafer, Connor Aylesworth, Benjamin L. King

**Affiliations:** 1Department of Molecular and Biomedical Sciences, University of Maine, Orono, ME 04469, USA; brandylee.dennis@maine.edu (B.-L.S.); alec.ballinger@maine.edu (A.B.); mykayla.weinstein@maine.edu (M.W.); haley.foreman@maine.edu (H.F.); julianna.grampone@maine.edu (J.G.); samuel.weafer@maine.edu (S.W.); connor.aylesworth@maine.edu (C.A.); 2Graduate School of Biomedical Science and Engineering, University of Maine, Orono, ME 04469, USA

**Keywords:** influenza virus, innate immune response, host–pathogen interactions, zebrafish

## Abstract

Influenza virus infection can cause severe respiratory disease and is estimated to cause millions of illnesses annually. Studies on the contribution of the innate immune response to influenza A virus (IAV) to viral pathogenesis may yield new antiviral strategies. Zebrafish larvae are useful models for studying the innate immune response to pathogens, including IAV, in vivo. Here, we demonstrate how Color-flu, four fluorescent IAV strains originally developed for mice, can be used to study the host response to infection by simultaneously monitoring infected cells, neutrophils, and macrophages in vivo. Using this model, we show how the angiotensin-converting enzyme inhibitor, ramipril, and mitophagy inhibitor, MDIVI-1, improved survival, decreased viral burden, and improved the respiratory burst response to IAV infection. The Color-flu zebrafish larvae model of IAV infection is complementary to other models where the dynamics of infection and the response of innate immune cells can be visualized in a transparent host in vivo.

## 1. Introduction

Influenza A virus (IAV) infection can result in acute respiratory inflammation that requires hospital care, and in severe cases, can lead to death. An estimated 10–37 million influenza infections occur each year in the US, with 114,000–624,000 needing hospital care for associated respiratory and heart symptoms [1]. Between 5000 and 27,000 deaths occur each year in the US from influenza infections [2]. While vaccines have been widely used, they have only been between 19 and 60% effective in preventing influenza because of antigenic variation in strains circulating in different populations and the difficulty of formulating vaccines against those strains [3,4]. Outbreaks of new influenza strains can become pandemic, such as the 2009 A(H1N1)pdm09 strain that resulted in an estimated 60.8 million infections, 274,000 hospitalizations, and 12,000 deaths in the US [5]. Antiviral therapies, such as oseltamivir (Tamiflu^®^), are valuable tools for treating influenza, but the potential risks that drug-resistant strains pose make it necessary to develop new therapies that target alternative mechanisms.

One roadblock to understanding the pathogenesis of influenza infection is the fact that the relative contributions of the virus and host factors have not been well characterized in vivo. Mammalian influenza virus infections originate in respiratory epithelial cells and alveolar macrophages [6]. The innate immune response to influenza infection includes type I interferons (IFNs) and proinflammatory cytokines and chemokines that activate neutrophils and macrophages. The innate antiviral immune response is activated in response to influenza infection and initiates a feed-forward loop that leads to the recruitment of excess neutrophils characterized with a dysregulation of IFN expression and reactive oxidative species (ROS) production [7]. Characterizing the complex dynamics of host immune cells during their response to influenza virus infection requires simultaneous monitoring of multiple cell types along with the virus. Biopsy studies in human, primate, or mammalian models cannot show the temporal dynamics of viral invasion and subsequent neutrophil and macrophage recruitment.

Influenza viruses that express fluorescent proteins are powerful tools for understanding viral pathogenesis in vivo. For example, Manicassamy et al. created a green fluorescent protein (GFP) reporter strain of A/Puerto Rico/8/34 (PR8; H1N1) that was used to study antigen presentation during IAV infection [8]. Fukuyama et al. generated four different mouse-adapted PR8 strains that express different fluorescent reporter proteins [9]. These “Color-flu” strains express either Venus (mVenus-PR8), a GFP with improved chromophore formation and brightness, enhanced cyan fluorescent protein (eCFP-PR8), enhanced GFP (eGFP-PR8), or mCherry (mCherry-PR8). For each strain, coding sequences for the fluorescent tags were fused to the end of the open reading frame for non-structural protein 1 (NS1) in the PR8 genome, allowing for the expression of NS1 fluorescent chimeric proteins during viral replication. Importantly, the development of these mouse-adapted strains included serial passaging and selecting for strains that had an increased pathogenicity and strong fluorescent protein expression to overcome the attenuation observed with viruses expressing reporter proteins [10,11]. Moreover, Color-flu strains were used to detect virus-infected cells in mouse bronchial tissue via imaging fluorescence in [9]. Simultaneous co-infections with the four strains allowed for areas of local virus propagation in the bronchial epithelium using multispectral imaging.

Zebrafish are powerful animal models for studying host responses to virus infection, as there are several genetic and pharmacological approaches that can be used to screen pathways and genes in combination with in vivo imaging of transparent embryos [12]. Major human immune signaling pathways that respond to viral infection are conserved in zebrafish [12]. Gabor et al. established the zebrafish model of IAV infection [13]. In their study, it was demonstrated that: (1) zebrafish embryos express α-2,6-linked sialic acid receptors; (2) embryos have reduced survival following IAV disseminated infection; (3) IAV replicates in embryos; (4) interferon phi 1 (*ifnphi1*) and myxovirus (influenza) resistance A (*mxa*) had upregulated expression with IAV infection; (5) disseminated IAV infection resulted in necrosis of the liver, gills, and head kidney tissue along with pericardial edema; and (6) pathological phenotypes from IAV infection were reduced in embryos treated with the neuraminidase inhibitor Zanamivir. Their study also showed how neutrophils are recruited to the site of localized infection in the swimbladder using fluorescent confocal imaging of zebrafish embryos infected with NS1-GFP PR8 IAV [8]. These initial studies demonstrated that in vivo imaging of fluorescent influenza strains in a vertebrate is possible. Goody et al. used the zebrafish IAV model to study how IAV infection exacerbated skeletal muscle damage in *sapje* zebrafish mutants [14]. The zebrafish IAV model was also recently used to demonstrate how cetylpyridinium chloride exposure increased survival and decreased viral burden following IAV infection compared to controls [15].

In this study, we demonstrate the application of Color-flu in zebrafish embryos to study host responses to IAV infection in vivo. We first compared the survival and viral burden of embryos to systemic infection by H1N1 PR8 with Color-flu. Next, we show how microinjecting Color-flu into the circulatory system results in disseminated infection throughout the embryo. We also show how Color-flu can be simultaneously imaged with neutrophil and macrophage reporter lines. As genetic background can influence phenotypes, we examined the survival of two wild-type zebrafish strains, AB and Ekkwill (EK), along with the pigment mutant, *casper* (*mitfa^w2/w2^; mpv17^a9/a9^*) [16,17], with disseminated infection. Next, we demonstrate how Color-flu can be used to test for the efficacy of two small molecules, ramipril and mitochondrial division inhibitor 1 (MDIVI-1), which were found to increase survival following systemic infection. Ramipril is an inhibitor of the angiotensin-converting enzyme (ACE) [18], and a recent study found that individuals who have prescriptions for ACE inhibitors had a lower risk of influenza [19]. MDIVI-1 inhibits dynamin 1-like protein (DNM1L) and blocks apoptosis by preventing mitochondrial and peroxisomal division [20]. Together, these studies demonstrate the utility of using Color-flu in a zebrafish model of IAV infection to study the host response and screen small molecules in vivo.

## 2. Materials and Methods

### 2.1. Zebrafish Care and Maintenance

The zebrafish used in this study were housed and maintained in the Zebrafish Facility at the University of Maine in accordance with the recommendations and standards in the *Guide for the Care and Use of Laboratory Animals* of the National Institutes of Health and the Institutional Animal Care and Use Committee (IACUC) at the University of Maine. Protocols utilized in this study were approved by the IACUC at the University of Maine (Protocol Number: A2021-02-02). Zebrafish were housed in recirculating tanks following standard procedures of a 14 h light, 10 h dark cycle at 28 °C [21]. The zebrafish lines used in this study were AB, Ekkwill (EK), *casper* (*mitfa^w2/w2^; mpv17^a9/a9^*) [16], and Tg(*mpeg1*:eGFP;*lyz*:dsRed). The Tg(*mpeg1*:eGFP;*lyz*:dsRed) line was created by crossing the Tg(*mpeg1*:eGFP) [22] and Tg(*lyz*:dsRed) [23] zebrafish lines. Embryos were obtained by spawning adult zebrafish using varying sets of females. Embryos were kept at 33 °C in 50 mL of sterilized egg water (60 μg/mL Instant Ocean Sea Salts; Aquarium Systems, Mentor, OH, USA) in 100 mm × 25 mm Petri dishes (catalog number 89220-696, VWR, Radnor, PA, USA), with water changes every 2 days.

### 2.2. MDCK/London Cell Culture

Madin–Dardy canine kidney/London (MDCK/London; Influenza Reagent Resource) cells (passage 3–4) were cultured using a modified protocol originally developed by Eisfield et al. [24]. Cells were grown at 37 °C with 5% CO_2_ in T-175 flasks (CELLSTAR Flasks; USA Scientific, Ocala, FL, USA), in minimal essential medium (MEM; catalog number 11090073, Gibco, Thermo Fisher Scientific, Waltham, MA, USA) containing final percentages/concentrations as follows: 5% heat-inactivated newborn calf serum (NCS; catalog number 26010074, Gibco), 2% heat-inactivated fetal bovine serum (FBS; catalog number 16140071, Gibco), 0.23% sodium bicarbonate solution (catalog number 25080094, Gibco), 2% MEM amino acids (from 50× stock; catalog number 11130051, Gibco), 1% MEM vitamin solution (from 100× stock; catalog number 11120052, Gibco), 4 mM L-glutamine (catalog number 25030081, Gibco), and 1% antibiotic–antimycotic (from 100× stock; catalog number 15240062, Gibco). The cells were maintained by washing twice with 1× Dulbecco’s phosphate-buffered saline (PBS, pH 7.4), trypsinized with 0.25% trypsin-EDTA with phenol red (catalog number 25300054, Gibco), and passaged in a 1:10 dilution every 2–3 days up to passage eight. Virus-infected cells were grown in MEM-BSA-TPCK media [24], which were prepared similarly to the MEM media described above but supplemented with 1 µg/mL Tosyl phenylalanyl chloromethyl ketone (TPCK) trypsin (Worthington Chemical Corporation, Lakewood, NJ, USA) and Bovine Albumin Fraction V (7.5% solution; catalog number 15260037, Gibco) instead of NCS and FBS.

### 2.3. Influenza Virus

PR8 influenza virus (A/PR/8/34 (H1N1) Purified Antigen; catalog number 10100374) was purchased from Charles River Laboratories (now AVS Bio in Norwich, CT, USA). Upon arrival, the virus was defrosted on ice, aliquoted into microcentrifuge tubes, and stored at −80°. Prior to use, virus aliquots were thawed on ice and diluted in cold sterile Hank’s buffered salt solution (HBSS) using a ratio of 87% virus to 13% diluent.

Color-flu [9] influenza virus strains MA-mVenus-PR8 (mVenus-PR8), MA-eCFP-PR8 (eCFP-PR8), MA-eGFP-PR8 (eGFP-PR8), and MA-mCherry-PR8 (mCherry-PR8) were kindly provided by Dr. Yoshihiro Kawaoka’s laboratory and stored at −80 °C. Color-flu virus strains were propagated in separated T-25 flasks (CELLSTAR Flasks; USA Scientific) using MDCK/London cells using MEM-BSA-TPCK as outlined in Eisfield et al. [24] (see MDCK/London Cell Culture section). Color-flu strains were grown for 4 days before being collected, filter sterilized using 0.45 µm tube top vacuum filters (VWR), aliquoted, and stored at −80 °C until use.

### 2.4. Microinjection

Microinjection was used to inject either vehicle (HBSS) controls or influenza virus to introduce a disseminated or localized infection in our zebrafish larvae. First, influenza virus strains were thawed on ice for 30 min. The virus strains were diluted in cold, sterile HBSS (catalog number 14170120, Gibco) at a solution of 87% virus and 13% HBSS. Next, the larvae were anesthetized in sodium bicarbonate-buffered MS-222 (tricaine methanesulfonate) solution (200 mg/L) (Syndel, Ferndale, WA, USA) at 2 or 3 dpf for disseminated infection experiments, or 4 dpf for localized infection experiments. The larvae were then lined up on a 2% agarose gel in a Petri dish coated with 3% methylcellulose. Microinjections of the virus or HBSS were conducted using pulled microcapillary needles (1.2 mm outside diameter, 0.94 mm inside diameter; Sutter Instruments, Novato, CA, USA) controlled with an MPPI-3 pressure microinjector (Applied Scientific Instruments, Eugene, OR, USA). For disseminated infection experiments of PR8 virus, 1 or 2 nL of virus (~2.8 × 10^8^ EID_50_ for lot 1, ~4.4 × 10^7^ EID_50_ for lot 2, ~1.7 × 10^6^ EID_50_ for lot 3) or HBSS was injected into the duct of Cuvier (DC) at 2 or 3 dpf, respectively. A total of 6 nL of Color-flu virus (~7.44 × 10^5^ TCID_50_/mL for mVenus-PR8, ~5.04 × 10^5^ TCID_50_/mL for eCFP-PR8, ~4.80 × 10^5^ TCID_50_/mL for eGFP-PR8, and ~4.56 × 10^5^ TCID_50_/mL for mCherry-PR8) or HBSS was injected into the DC at 3 dpf for disseminated Color-flu infection experiments. For localized infection experiments with PR8 virus, 4 nL (~4.4 × 10^7^ EID_50_ for lot 2, ~1.7 × 10^6^ EID_50_ for lot 3) of virus or HBSS was injected into the swimbladder of the larvae at 4 dpf. For localized Color-flu infection experiments, 10 nL of Color-flu or HBSS was injected into the swimbladder at 4 dpf. For the control experiments, the larvae were also injected with heat-inactivated or UV-inactivated virus. Virus aliquots were either heat-inactivated at 80 °C for 5 min, or UV-inactivated with 254 nm light exposure for 60 min on ice (UV CrossLinker, VWR). For virus infections, microcapillary needles were changed hourly to keep the virus viable. For disseminated infection experiments, zebrafish were sorted into Petri dishes at a density of ~50 larvae/dish and maintained in embryo water at 33 °C (see Zebrafish Care and Maintenance section). For localized infection experiments, zebrafish were kept at a density of ~45 larvae/dish.

### 2.5. Drug Exposures

For all of our small molecule drug studies, virus-infected or HBSS-injected larvae were exposed to either dimethyl sulfoxide (DMSO; Sigma-Aldrich, St. Louis, MO, USA), DMSO-solubilized ramipril (Cayman Chemical Company, Ann Arbor, MI, USA), or DMSO-solubilized MDIVI-1 (Cayman Chemical Company, Ann Arbor, MI, USA) by adding these solutions into the embryo water at 24 hpi. The larvae were exposed to DMSO, ramipril, or MDIVI-1 for one hour at 33 °C in a dark incubator. After exposure, zebrafish were transferred to 50 mL of fresh embryo water twice to rinse away the DMSO or DMSO-solubilized drugs. The final concentrations used were 0.2 nM for ramipril [25] and 7 nM for MDIVI-1 [26].

### 2.6. Survival Studies

For survival studies, mortality was monitored and counted daily in infected and control-injected larvae for up to 7 dpf. Larvae were maintained in dishes (~50 larvae/dish) at 33 °C with embryo water changes every two days.

### 2.7. Viral Burden Assays

Tissue culture infectious dose 50 (TCID_50_) end-point dilution assays using MDCK/London cells were used to measure the viral burden of influenza virus in zebrafish. Cohorts of 200 zebrafish per experimental group were used (see Zebrafish Care and Maintenance section), infected (see Microinjections section), and maintained in embryo water at 33 °C in dishes (~50 larvae/dish) with water changes every other day. Zebrafish were collected at 0, 24, 48, 72 and 96 hpi for the larvae infected at 2 dpf, and 0, 24, 48 and 72 hpi for the larvae infected at 3 dpf. At the appropriate timepoint, 25 larvae per group were collected and euthanized with an overdose (300 mg/L) of MS-222 for 10 min. Next, the larvae were transferred into 500 µL of RNAlater (Invitrogen, Thermo Fisher Scientific), flash-frozen with liquid nitrogen, and stored at −80 °C.

The MDCK/London cells were plated into 96-well plates (PlateOne; catalog number 1837-9600, USA Scientific) the evening before the TCID_50_ assay at a cell density of ~15,000 cells per well to achieve 90–95% confluency the next day. After thawing the frozen samples on ice, the RNAlater was replaced with 500 µL of MEM-BSA-TPCK (see MDCK/London Cell Culture section). Samples were homogenized with a Bullet Blender tissue homogenizer (Next Advance, Troy, NY, USA) using a sterile metal bead at setting #3 for 5 min at 4 °C and centrifuged at 8000× *g* for 1 min. Next, eight 1-to-8.5 serial dilutions (10-0.9 to 10-7.4) for each sample were prepared in MEM-BSA-TPCK. Cells were washed twice with PBS prior to adding the serial dilutions. After removing the second PBS wash, 50 µL serial dilutions for each sample were plated using triplicate wells (24 wells/sample), with 4 control wells per plate; 50 µL of MEM-BSA-TPCK was added to the control wells. Plates were centrifuged at 2000× *g* for 10 min at 4 °C and then incubated at 37 °C for two hours with 5% CO_2_. Cell media were removed from all wells and 105 µL of MEM-BSA-TPCK was added to each well. Plates were then incubated at 37 °C for 72 h with 5% CO_2_. Next, cytopathic effects were observed, and the cells were counted using a Bio-Rad TC20 Automated Cell Counter (Bio-Rad, Hercules, CA, USA). TCID_50_/mL was calculated using the Spearman–Kärber method [27].

### 2.8. Respiratory Burst Assays

The capacity of zebrafish to generate ROS in vivo was quantified using a respiratory burst assay. Virus-infected and HBSS-injected zebrafish were collected at 24 and 48 hpi, anesthetized in sodium bicarbonate-buffered MS-222 (200 mg/L), and placed into black, flat-bottom 96-well plates (Fluotrac 600, Greiner-Bio., Monroe, NC, USA) with 100 µL of embryo water. The HBSS-injected zebrafish were placed into the wells in the first 6 columns of the plate, and the virus-infected zebrafish were placed into the remaining wells. The first 2 columns were treated with 5 µL of 1 mM protein kinase C inhibitor, bisindolylmaleimide I (BisI, Cayman Chemical Company), in DMSO. The BisI-treated embryos were incubated for 30 min at 28 °C. Zebrafish in columns 1, 3–4, and 7–9 were treated with 100 µL of embryo water plus 1 µg/mL 2′,7′-dichlorofluorescein diacetate (H2DCFDA) (Sigma-Aldrich) in 0.4% DMSO. Zebrafish in columns 2, 5–6, and 10–12 were treated with 100 µL of embryo water plus 1 µg/mL H2DCFDA and 400 ng/mL phorbol 12-myristate 13-acetate (PMA, Sigma-Aldrich). The plates were then covered with aluminum foil and incubated at 28 °C for 2.5 h. Fluorescence was read using a plate reader (BioTek Synergy, Agilent Technologies, Santa Clara, CA, USA). Assays were repeated three times.

### 2.9. Confocal Imaging

Zebrafish were anesthetized in sodium bicarbonate-buffered MS-222 (200 mg/L) and placed in 24-well glass-bottom imaging plates (MatTek, Ashland, MA, USA) with embryo water with 0.7% agarose. An Olympus Fluoview IX-81 inverted microscope with an FV1000 confocal system with 405, 458, 488, 514, and 543 nm laser lines was used for fluorescence and brightfield imaging of the zebrafish. Images were obtained using ×4 or ×10 objectives. Ten zebrafish were scanned per group and one was randomly chosen to visualize Color-flu infected AB larvae. Twelve zebrafish were scanned per group and one was randomly selected to visualize eCFP-PR8 infected Tg(*mpeg1*:eGFP;*lyz*:dsRed) larvae. For quantification of the number of neutrophils, macrophages, and relative abundance of virus infected cells, six Tg(*mpeg1*:eGFP;*lyz*:dsRed) larvae were scanned over two separate trials, and the four healthiest zebrafish from each experiment were selected for analysis. The Z-stack cross sections were 5 microns thick, and five slices were imaged both dorsally and ventrally (50 microns total) from the center of the larvae and then analyzed.

### 2.10. Image Analysis

Image analyses to count neutrophils and macrophages and quantify virus levels based on fluorescence intensities were conducted using MATLAB (version R2023a; The MathWorks Inc., Natick, MA, USA). Longitudinal images obtained on the confocal were composed of 5 µm-thin sections. Five sections proximal to the center of the zebrafish and five sections distal to the center of the zebrafish were analyzed for a total of 50 µm (approximately 40–50% of the total zebrafish width). Masks were generated to identify dsRed-tagged neutrophils, eGFP-tagged macrophages, and eCFP-PR8 Color-flu virus. Those masks were then used to count both neutrophils and macrophages, and the level of eCFP-PR8 virus infection (pixels).

### 2.11. qRT-PCR Assays

Four biological replicate total RNA samples were prepared per sample group using 8 larvae per sample. Each set of larvae for a given sample was homogenized using a Next Advance Bullet Blender (Next Advance, Troy, NY, USA) with a single 5 mm sterile, stainless steel metal bead in 360 μL of Trizol (Invitrogen, Waltham, MA, USA) for 3 min. Homogenates were centrifuged for 3 min at 8000× *g* and transferred to new tubes. Total RNA was extracted using the Direct-zol RNA microprep kit (Zymo Research, Irvine, CA, USA) following the manufacturer’s protocol. cDNA was synthesized using the ProtoScript II First Strand cDNA synthesis kit (New England Biolabs, Ipswich, MA, USA), following the manufacturer’s protocol. qRT-PCR assays were conducted using Bio-Rad SsoAdvance Universal SYBR Green Mastermix (Bio-Rad, Hercules, CA, USA), and oligos (IDT, Coralville, IA, USA), shown in Appendix A, and a Bio-Rad CFX96 instrument (Bio-Rad) according to the manufacturer’s protocol.

### 2.12. Statistical Analysis

GraphPad Prism 9.5.1 (GraphPad Software, Boston, MA, USA) was used to generate and analyze survival curves and graphs. The Kaplan–Meier survival analysis method was used to analyze survival curves with 95% confidence intervals. Mantel–Cox test *p*-values <0.05 between the sample groups were considered significant. Two-way ANOVAs followed by Dunnett’s multiple comparison tests were used to analyze the Color-flu TCID_50_/mL values. A Brown–Forsythe one-way ANOVA test was used to analyze the TCID_50_/mL values from the live PR8, heat-inactivated and UV-inactivated larvae, followed by Dunnett’s multiple comparison tests. Statistical analyses of the fold induction from respiratory burst assays were conducted using the Kruskal–Wallis test with the Dunn’s multiple comparison test for pairwise comparisons. Pairwise comparisons with an adjusted *p*-value < 0.05 were considered significant.

## 3. Results

### 3.1. IAV Infection Decreases Survival and Replicates in Zebrafish

It has previously been shown that zebrafish express α-2,6-linked sialic acid-containing receptors and are susceptible to infection by PR8 H1N1 and X-31 A/Aichi/68 H3N2 IAV [13]. We modified the original zebrafish IAV infection protocol to achieve approximately 50% mortality by 7 days post fertilization (dpf) by infecting embryos with PR8 IAV at either 2 or 3 dpi, by increasing the level of virus infection compared to the original protocol, and by using Hank’s buffered salt solution (HBSS) instead of phosphate-buffered saline (PBS) as the diluent. In our protocol, we microinjected a 1 nL solution of 87% PR8 IAV and 13% HBSS (~1.7 × 10^6^ to ~2.8 × 10^8^ EID_50_, depending on the virus lot) into the duct of Cuvier (DC) in 2 or 3 dpf anesthetized embryos. This virus concentration was higher than the original study, which used 1.5 nL (~5 × 10^3^ EID_50_) of PR8. We found reduced survival in the AB zebrafish systemically infected with three different PR8 virus lots at 2 dpf compared to vehicle (HBSS) controls (Figure 1A). The percentage of survival after 5 days was 57.6%, 55.6%, and 57.6% following infection with the three lots. Similar reductions in survival were observed in zebrafish systematically infected with two different PR8 lots at 3 dpf compared to vehicle controls (Figure 1B). After 4 days, 55.3% and 56.4% of the larvae survived following injection by the two lots. Injection of heat- and ultraviolet (UV)-inactivated PR8 in 2 dpf AB zebrafish did not alter survival compared to vehicle controls (Appendix A). Increased viral titer was observed at 24 hpi in 2 dpf infected AB zebrafish, but not in heat- and UV-inactivated PR8 (Appendix A).

### 3.2. Color-Flu Infection Decreases Survival and Replicates in Zebrafish

Using Color-flu [9] stocks, we examined survival and viral burden in systemically infected zebrafish at 3 dpf (Figure 1C,D and Appendix A). Due to the lower virulence of Color-flu [9], our infection protocol was modified to inject a 6 nL solution of 87% Color-flu (~7.44 × 10^5^ TCID_50_/mL for mVenus-PR8, ~5.04 × 10^5^ TCID_50_/mL for eCFP-PR8, ~4.80 × 10^5^ TCID_50_/mL for eGFP-PR8, and ~4.56 × 10^5^ TCID_50_/mL for mCherry-PR8) and 13% HBSS into the DC, and 6 nL of HBSS for controls. Decreased survival was observed for all four of the strains (Appendix A). Consistent with studies on these Color-flu strains in mice [9], we observed higher survival rates for zebrafish infected with Color-flu than those infected with PR8. The larvae survival rate after 4 days was lowest for the mVenus-PR8 (66.7%) and eCFP-PR8 (69.6%) strains, and higher for the mCherry-PR8 (80.0%) and eGFP-PR8 (80.8%) strains. Consistent with the virulence shown in the survival studies, the zebrafish systemically infected with Venus-PR8 and eCFP-PR8 had higher viral titers at 0 hpi than the other two Color-flu strains (Figure 1D). Significant increases in viral titers were observed at 24 hpi and 48 hpi for all Color-flu strains.

### 3.3. Color-Flu Infection Induces Proinflammatory Gene Expression

We examined the expression of six proinflammatory genes in response to PR8 and mVenus-PR8 infection at 24 hpi (Appendix A). These genes are known to be activated by major inflammatory response pathways, including type I interferon, Toll-like receptor (TLR), and cytokine signaling, as well as ROS production. IAV infection has previously been shown to induce a type I interferon response in zebrafish larvae [13] that, in turn, induces inflammation. Interferon regulatory factor 9 (*irf9*) activates the type I interferon response during viral infection [28], and this was upregulated with mVenus-PR8 infection. Humans with mutations in *IRF9* have the immunologic disorder Immunodeficiencey-65, and are susceptible to viral infections [28,29]. Zebrafish receptor (TNFRSF)-interacting serine-threonine kinase 1 (*ripk1l*) is orthologous to human RIPK1, which participates in Toll-like receptor (TLR) and retinoic acid-inducible gene I (RIG-I) signaling following viral infection [30] and which was upregulated with mVenus-PR8 infection. The suppressor of cytokine signaling 3b (*socs3b*) is the zebrafish ortholog of human *SOCS3* and is a negative regulator of cytokine signaling that has been shown to be upregulated following influenza virus infection, resulting in an overexpression of interleukin 6 [31]. The expression of neutrophil cytosolic factor 1 (*ncf1*), which encodes a subunit of NADPH oxidase, was decreased at 24 hpi with mVenus-PR8 infection (Appendix A).

### 3.4. Zebrafish Lines Respond Differently to Influenza Infection

Multiple zebrafish lines have been used to model a wild-type response to injury and infection. To evaluate the consistency of the zebrafish response to IAV infection, we compared the response of three lines to infection: AB, the wild line; EkkWill (EK); and the pigmentation mutant, *casper* (*mitfa^w2/w2^; mpv17^a9/a9^*). The AB line was originally used to establish the zebrafish as a model for studying the innate immune response to influenza virus [13] as well as bacterial [32] and fungal infection [33]. The EK line has been used to study fin [34] and cardiac [35] tissue regeneration. The *casper* line is optically transparent throughout development into adulthood, allowing for various studies, including research on stem cells and tumor biology [16]. Systemic PR8 infection in 2 dpf embryos resulted in reduced survival for all three lines compared to HBSS vehicle controls (Figure 2A). *Casper* larvae had the lowest percentage of survival (44.2%) after 5 days, followed by EK (63.0%) and AB (62.6%). For the embryos infected with PR8 at 3 dpf and followed for 4 days, *casper* larvae also had the lowest survival rate (45.7%) compared with EK (53.7%) and AB (62.7%) (Figure 2B).

Viral load was measured at daily intervals following the systemic PR8 infection of 2 dpf embryos from all three lines using tissue culture infectious dose (TCID_50_) virus titer assays (Figure 2C). Viral titers increased by 24 hpi in all three lines, peaked at 48–72 hpi, and then declined by 96 hpi. The AB larvae had their peak viral load at 72 hpi and still had an increased load at 96 hpi. The EK and *casper* larvae had their peak viral loads at 48 hpi, with levels not dissimilar to the 0 hpi controls at 96 hpi. Viral load also increased with systemic PR8 infection in 3 dpf embryos from the three lines by 72 hpi (Figure 2D). The *casper* larvae had their peak viral load at 48 hpi, like the 2 dpf infected embryos.

Localized PR8 infection in the swimbladder of 4 dpf AB, EK, and *casper* larvae also resulted in decreased survival by 3 dpi over the HBSS controls (Appendix AA). Like the survival observed with systemic infection, PR8-infected *casper* larvae also had the lowest survival rate (31.5%) compared to EK (54.5%) and AB (61.9%). Likewise, localized Color-flu (mVenus-PR8) infection in the swimbladder of 4 dpf AB, EK, and *casper* larvae also resulted in reduced survival, with *casper* having the lowest percentage of survival (42.3%), followed by EK (67.5%) and AB (74.1%) (Appendix AB).

### 3.5. Live Confocal Imaging of Zebrafish Infected with Color-Flu

The optical transparency of zebrafish embryos and larvae allows for in vivo confocal imaging of Color-flu infection, where the host response can be visualized. With disseminated infection, virus-infected cells were observed throughout the AB larvae for each of the four Color-flu strains (Figure 3). In these whole larvae lateral views at 24 hpi, the highest density of viral infection was in the yolk sac and yolk sac extension. Virus-infected cells were also observed in the skeletal muscle. Several zebrafish transgenic fluorescent reporter strains have been used to visualize cell types, including neutrophils and macrophages. We injected the dual macrophage and neutrophil reporter line Tg(*mpeg1*:eGFP;*lyz*:dsRed) with eCFP-PR8 and a vehicle (HBSS) control at 3 dpf and imaged the larvae at 24 hpi (Figure 3B). The control larvae show macrophages and neutrophils in their circulatory system and other tissues including the skeletal muscle. Imaging of the eCFP-PR8-infected larvae allows for the simultaneous visualization of macrophages, neutrophils, and infected cells.

### 3.6. Evaluating Small Molecules That Alter the Response to IAV Infection

Next, we evaluated how two small molecule drugs, ramipril and MDIVI-1, would alter the response to IAV infection. We examined survival, viral burden, and immune capacity via respiratory burst assays. We also used confocal imaging to study the relative abundance of neutrophils and macrophages, and the level of viral infection. To model drug therapies administered after infection, embryos were infected with IAV at 3 dpf, then treated with either DMSO (control), ramipril, or MDIVI-1 at 24 hpi for one hour, and then washed and maintained in clean embryo water for up to 4 days post infection. The ramipril and MDIVI-1 administered to the PR8-infected and vehicle (HBSS) control larvae were at concentrations of 0.2 nM and 7nM, respectively, which increased survival (Figure 4). Ramipril increased survival to 90.0% compared to 52.1% in the DMSO controls. MDIVI-1 increased survival to 85.4% compared to 55.4% in DMSO controls. With mVenus-PR8-infected larvae, ramipril or MDIVI-1 treatment increased survival to 87.7% and 85.0%, respectively, compared to 58.9% in DMSO controls. The ramipril and MDIVI-1 concentrations used were selected after evaluating differences in survival due to drug treatment in PR8-infected larvae at 2 and 3 dpf (Appendix A). The range of concentrations evaluated for ramipril were 0.1, 0.2, 0.3, and 0.4 nM. For MDIVI-1, the range of concentrations evaluated were 3, 5, 7, and 10 nM. Both the 0.1 and 0.2 ramipril treatments, and only the 7 nM MDIVI-1 treatment, increased survival in larvae infected with PR8 at both 2 and 3 dpf. The highest doses of ramipril (0.4 nM) and MDIVI-1 (10 nM) both decreased survival in the control (HBSS) injected larvae.

The ramipril and MDIVI-1 treatments also reduced the viral burden in IAV-infected larvae by 24 h after treatment (Figure 5). In the larvae infected with PR8 at 2 and 3 dpf, the viral burden increased at 24 hpi when the larvae were treated with ramipril, MDIVI-1, or DMSO. By 48 hpi, the viral burden was reduced in ramipril- and MDIVI-1-treated larvae. In the 2 dpf larvae infected with ramipril and MDIVI-1 treatment, the viral burden at 72 hpi was reduced to levels measured at 0 hpi. A reduction in viral burden following ramipril and MDIVI-1 treatment was also observed at 48 hpi with mVenus-PR8 infection (Figure 5C).

Next, the level of ROS generated by the larvae following infection was quantified in order to characterize the capacity of the immune system to mount a respiratory burst response. In PR8- and mVenus-PR8-infected larvae, the respiratory burst response was reduced compared to uninfected controls (Figure 6). Ramipril and MDIVI-1 treatment rescued the response, such that the level of induction was the same or higher than that of the uninfected controls.

We then characterized the abundance of neutrophils and macrophages and the level of viral infection using fluorescent confocal imaging of Tg(*mpeg1*:eGFP;*lyz*:dsRed) larvae infected with eCFP-PR8. In DMSO-treated larvae, we observed higher numbers of neutrophils and viral infection, but lower numbers of macrophages at 48 hpi (Figure 7). Ramipril increased the number of neutrophils in infected larvae compared to uninfected controls, a difference not observed with the macrophages. With eCFP-PR8 infection, the number of macrophages increased with MDIVI-1 treatment over DMSO controls. The level of eCFP-PR8 infection was reduced with ramipril and MDIVI-1 treatment compared to DMSO controls.

Time-lapse in vivo confocal imaging allows cells to be tracked in these transgenic lines injected with Color-flu and then exposed to small molecules. Time-lapse imaging showed circulating neutrophils and macrophages over the course of 30 min in 5 dpf larvae that were injected with HBSS at 3 dpf (Appendix A). In 5 dpf larvae infected with eCFP-PR8 at 3 dpf and then exposed to DMSO at 24 hpi for one hour, we observed viral infection in several tissues (Appendix A). We observed macrophages and neutrophils near the highest density of viral infection, and more circulating macrophages over the course of 30 min. In eCFP-PR8-infected larvae exposed to ramipril, we observed lower viral infection, more neutrophils, and fewer macrophages over 30 min (Appendix A). The number of macrophages was increased in eCFP-PR8-infected larvae exposed to MDIVI-1 over 30 min (Appendix A).

## 4. Discussion

Here, we demonstrate that Color-flu can be used to model the innate immune response to IAV infection in zebrafish larvae, and that both ramipril and MDIVI-1 treatment improve survival and reduce viral burden following IAV infection. There is an urgent need to develop new antiviral therapies for IAV due to the threat that new IAV strains will pose in the future. The IAV strains that caused the last four influenza pandemics since 1900 were derived from the viral transmission of strains from avian and non-human mammalian hosts and their subsequent recombination with human strains [36]. Furthermore, antiviral therapies can become ineffective when the right combination of mutations occur in the virus. For example, a single amino acid substitution (H274Y) in neuraminidase was reported in H5N1 isolates that conferred resistance to oseltamivir [37]. The zebrafish model of IAV infection is complementary to other animal and cell line models used to evaluate antiviral therapies. We show how the model can be used to evaluate differences in survival, viral burden, respiratory burst, gene expression, and the dynamics of the neutrophil and macrophage response. This zebrafish Color-flu model of IAV infection is unique as it is the only model where innate immune and virus-infected cells can be visualized in a transparent host in vivo. As zebrafish larvae only have innate immune cells at this stage of development, the model can be used to study the roles of neutrophils and macrophages during IAV infection. Studies of the innate immune response to IAV are needed as the virus can evade the innate response through interactions of viral proteins (PB1, PB1-F2, PB2, PA, and NS1) with the host [38,39,40,41,42,43].

We demonstrate that zebrafish larvae are a robust model for analyzing the innate immune response to IAV infection by evaluating multiple lots of PR8 viruses and multiple zebrafish lines. The landmark paper that first established the zebrafish larvae as a model for IAV infection [13] had studied the AB line. Here, we demonstrate infection in two “wild-type” lines, AB and EK, along with the pigmentation mutant *casper*. When the same amount of virus is injected in larvae from these strains, *casper* had the lowest level of survival and highest viral burden. As is carried out in other model organisms, comparing strains with varying phenotypes can be a powerful way to understand disease mechanisms [44]. Whole genome sequencing of *casper*, a related pigmentation mutant, *roy*, and AB zebrafish revealed 4.3 million single-nucleotide polymorphisms between strains, including mutations in the *mpv7* and *mitfa* genes in the *casper* line [45]. In addition to cataloging sequence variants, the authors characterized gene expression in skin and skeletal muscle of adult *casper* and *roy* mutants compared to wild-type AB zebrafish [45]. The interferon response stimulator *sting1* was consistently upregulated in the skin and skeletal muscle of *casper* mutants [45]. The higher expression of *sting1*, along with any of the 11,583 non-synonymous variants found in *casper* [45], could potentially explain the difference in response to IAV infection.

The capability to simultaneously visualize both virus-infected cells and fluorescently tagged host cells, such as innate immune cells, in transparent living larvae makes the zebrafish Color-flu model unique. Zebrafish larvae from fluorescent reporter strains that label neutrophils and/or macrophages with green or red fluorescent reporters have enabled studies of host–pathogen interactions, such as those during bacterial and fungal infections. However, many of these studies have often included the use of a green or red fluorescently labeled pathogen, thereby preventing the simultaneous imaging of this pathogen with both host immune cell types. As we demonstrate here, the four Color-flu strains developed by the Kawaoka laboratory make it possible to image virus-infected cells along with neutrophils and macrophages using distinguishable color combinations. Furthermore, the Color-flu strains retained their virulence despite the introduction of the fluorescent transgene, as we demonstrate by showing the decreased survival and increased viral burden of infected AB, EK, and *casper* larvae.

We describe the response to both systemic and localized IAV infections. To study the systemic response, IAV was directly injected into the duct of Cuvier so that the virus could spread throughout the larvae via the circulatory system. Confocal imaging of Color-flu-infected larvae showed viral infection throughout different tissues of the larvae. The highest density of infected cells was in the yolk sac and yolk sac extension. The yolk sac accumulation of compounds has been noted in toxicological studies using zebrafish larvae [46]. In one study on the toxicity of nanoplastics, embryos were exposed to 70 nm diameter fluorescent polystyrene nanoparticles in embryo water [47]. By 1 dpf, fluorescence was detected in the yolk sac and yolk sac extension of 1 and 10 ppm exposed embryos, which persisted until 5 dpf, when the yolk sac is reabsorbed after the gastrointestinal system becomes active. As the diameter of nanoparticles is slightly smaller than IAV particles (80–120 nm), it is plausible that the virus particles accumulate in the yolk sac and yolk sac extension like these nanoparticles, thus resulting in an increase in the level of infection in these tissues.

We applied the zebrafish Color-flu model to evaluate whether the ACE inhibitor, ramipril, or the autophagy inhibitor MDIVI-1 would alter the response to IAV infection. Both small molecules improved the response to infection, as the treated larvae had increased survival, lower viral burden, and a respiratory burst response that was the same or higher than that of the uninfected controls. Quantification of fluorescent confocal images of Tg(*mpeg1*:eGFP;*lyz*:dsRed) larvae infected with Color-flu showed lower levels of viral infection in ramipril- and MDVI-1-treated larvae. Ramipril treatment increased the relative number of neutrophils in the infected larvae, whereas MDIVI-1 increased the relative number of macrophages.

Ramipril is an ACE inhibitor that is frequently prescribed for hypertension, as ACE produces angiotensin II, which, in turn, elevates blood pressure [48]. ACE has also been associated with innate immune function [49]. Angiotensin II mediates proinflammatory responses, including the production of reactive oxygen species (ROS) by NADPH oxidase and the subsequent activation of nuclear factor-κB (NF-κB) and activator protein 1 (AP1) [50]. An overexpression of ACE in mouse neutrophils increased superoxide production and enhanced the clearance of bacteria in mice infected with methicillin-resistant *Staphylococcus aureus* (MRSA), *Pseudomonas aeruginosa*, or *Klebsiella pneumoniae* [51]. ACE overexpression in mouse macrophages also lowered the bacterial burden in mice infected with MRSA or *Listeria monocytogenes* [52]. While ACE overexpression in myeloid cells benefit immune responses, ACE deficiency can also be beneficial. A study on mouse models of acute respiratory distress syndrome showed that ACE-deficient mice exhibit less lung damage [53]. Interestingly, a study of electronic health records of patients in the Clinical Practice Research Datalink in the United Kingdom showed a lower risk of influenza infection depending on the duration of ACE inhibitor use [19]. Our study of ramipril treatment in our zebrafish IAV model demonstrated an improved response to viral infection.

Mitochondria have important roles in immune function, including regulating inflammation [54]. Mitophagy maintains mitochondrial function by clearing mitochondria when they become damaged [55]. A myriad of biological processes is altered when mitochondrial function is disrupted, including the overproduction of mitochondrial ROS, which can activate the nucleotide oligomerization domain (NOD)-like receptor family pyrin domain-containing 3 (NLRP3) inflammasome [56]. IAV infection has been shown to induce mitophagy, thereby altering inflammasome activation [57,58]. The IAV nucleoprotein (NP) has been shown to induce mitophagy through the toll interacting protein (TOLLIP) and mitochondria antiviral signaling protein (MAVS) [59]. In that study, the mitophagy inhibitor MDIVI-1 was found to reduce NP-induced degradation of mitochondrial proteins. The mitochondrial fission inhibitor MDIVI-1 was also shown to reduce NLRP3 inflammasome activation in transmitochondrial cybrid cells with diminished mitochondrial function [60]. In our studies of IAV-infected zebrafish larvae, we show that inhibiting mitophagy using MDIVI-1 treatment increases survival and decreases viral burden. We can therefore hypothesize that MDVI-1 counters IAV-induced mitophagy and NLRP3 inflammasome activation.

Zebrafish embryos and larvae have been used to screen small molecule drugs where the response of separate larvae to different treatments can be monitored in multi-well plates [61]. Large-scale screening of small molecules that alter the innate immune response to IAV infection is feasible using our Color-flu model. Color-flu-infected larvae would be exposed to small molecules within wells, and then characterized using fluorescent confocal imaging to determine their relative levels of infection. In these studies, small molecules could be evaluated using different concentrations as well as in different combinations.

## 5. Conclusions

We demonstrate that Color-flu can be used to study the innate immune response to IAV infection in zebrafish larvae. This model is complementary to other models of IAV infection. Our model is the only IAV model where the dynamics of infection and the response of innate immune cells can be visualized in a transparent host in vivo. The roles of neutrophils and macrophages during IAV infection can also be readily characterized using this model. Using Color-flu, we characterize how ACE inhibition by ramipril and mitophagy inhibition by MDIVI-1 both improve the response to IAV infection by limiting inflammation through distinct mechanisms. These studies demonstrate how larger small molecule screens are possible using this zebrafish Color-flu model of IAV infection. Such screening studies are needed in order to find small molecules that could be developed into novel antiviral therapies for influenza viruses.

## Figures and Tables

**Figure 1 viruses-16-00155-f001:**
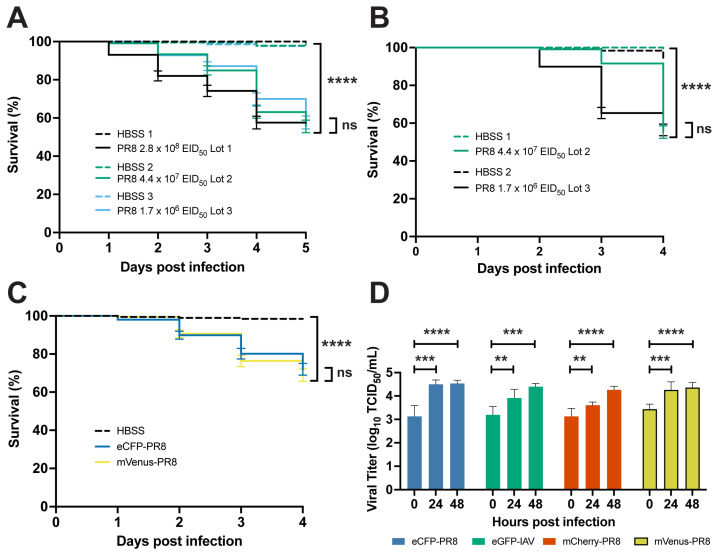
Characterization of PR8 and Color-flu systemically infected AB zebrafish. (**A**) Decreased survival of AB zebrafish systemically infected with three different lots of PR8 IAV at 2 dpf compared to vehicle (HBSS) controls (*p* < 0.0001 for each lot comparison). Survival rates of PR8-infected embryos were not significantly different between lots (*p* = 0.0603). (**B**) Decreased survival of AB zebrafish systemically infected with two different lots of PR8 IAV at 3 dpf compared to vehicle controls (*p* < 0.0001 for each lot comparison). Survival rates of PR8-infected embryos were not significantly different between lots (*p* = 0.1442). (**C**) Decreased survival of AB zebrafish systemically infected with eCFP-PR8 or Venus-PR8 (3.2 × 102 TCID_50_/mL) compared to vehicle controls (*p* < 0.0001 for each comparison). Survival rates of eCPF-PR8- or Venus-PR8-infected zebrafish were not significantly different strains (*p* = 0.5238). (**D**) Increased TCID_50_ viral titer in Color-flu systemically infected AB zebrafish at 24 and 48 hpi compared to 0 hpi. eCFP-PR8-, mCherry-PR8-, and Venus-PR8-infected zebrafish had increased viral titers at 24 and 48 hpi compared to 0 hpi (adjusted *p*-values = 0.0001 and <0.0001, respectively, for eCFP-PR8; 0.0019 and 0.0001, respectively, for eGFP-PR8; 0.0047 and <0.0001, respectively, for mCherry-PR8; and 0.0003 and <0.0001, respectively, for Venus-PR8). Survival studies were conducted with four independent experiments (*n* = 4) and 50 larvae per sample group. TCID_50_ assays were carried out with three independent experiments (*n* = 3) and 25 larvae per group for each time point. Not significant (ns), *p* > 0.05; ** *p* < 0.01; *** *p* < 0.001; **** *p* < 0.0001.

**Figure 2 viruses-16-00155-f002:**
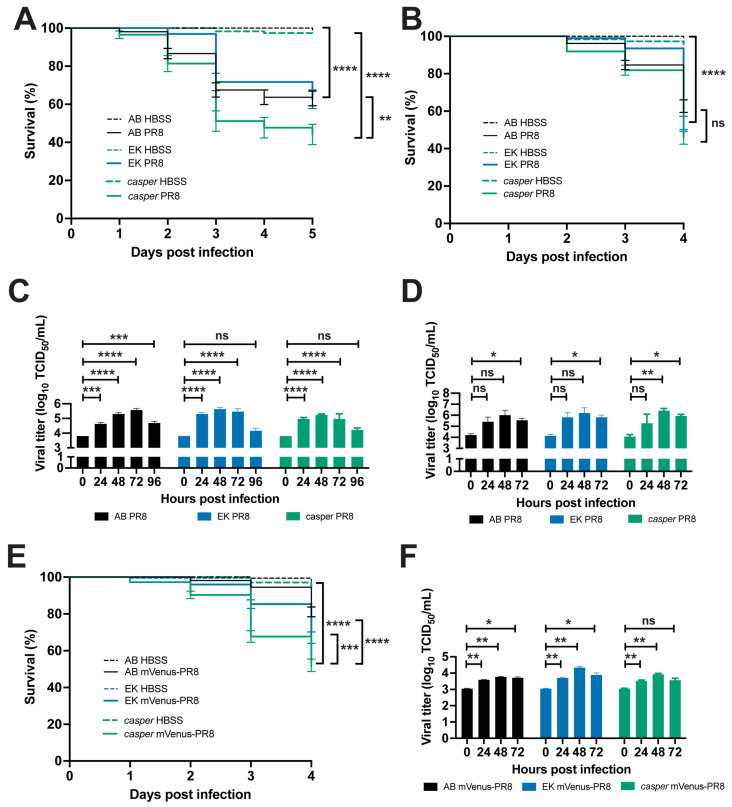
Characterization of PR8 and Color-flu systemic infection across zebrafish lines. (**A**) Decreased survival was greater in *casper* than AB or EK lines with 2 dpf systemic PR8 infection. PR8-infected zebrafish had a decreased survival rate compared to vehicle (HBSS) controls (*p* < 0.0001 for all lines). *Casper* zebrafish had a lower survival rate than EK (*p* = 0.0024) and AB (*p* = 0.0055) according to the log-rank Mantel–Cox test. (**B**) Decreased survival rates were observed in AB, EK, and *casper* with 3 dpf PR8 infection compared to controls (*p* < 0.0001 for all lines). No significant (ns) difference in survival rate was detected between PR8-infected AB, EK, and *casper* larvae. (**C**) Increased TCID_50_ viral titer in 2 dpf PR8-infected AB, EK, and *casper* zebrafish at 24, 48, and 72 hpi compared to 0 hpi (adj. *p*-value < 0.001 for all comparisons except for AB 24 hpi (adj. *p*-value = 0.0006) and AB 96 hpi (adj. *p*-value = 0.0003)). (**D**) Increased TCID_50_ viral titer in 3 dpf PR8-infected AB, EK, and *casper* zebrafish at 72 hpi compared to 0 hpi (adj. *p*-value = 0.0323, 0.0211, and 0.0452, respectively) and *casper* zebrafish at 48 hpi compared to 0 hpi (adj. *p*-value = 0.0063). (**E**) Decreased survival with 3 dpf mVenus-PR8-infected AB, EK, and *casper* zebrafish compared to HBSS controls (*p* < 0.0001 for all lines). As observed with PR8 infection, *casper* zebrafish had the lowest survival rate (67.7%), which was lower than that of AB (94.5%, *p* < 0.0001) and EK (85.5%, p < 0.0002). (**F**) mVenus-PR8-infected AB, EK, and *casper* zebrafish had an increased viral titer at 24 and 48 hpi compared to 0 hpi (adjusted *p*-values = 0.0012 and 0.0022, respectively, for AB; 0.0025 and 0.0027, respectively, for EK; and 0.0045 and 0.0066, respectively, for Venus-PR8). mVenus-PR8-infected AB and EK zebrafish had an increased viral titer at 72 hpi compared to 0 hpi (adjusted *p*-value = 0.0236, and 0.0281, respectively). Survival studies were conducted with four independent (*n* = 4) experiments and 50 larvae per sample group. TCID_50_ assays were carried out with three independent (*n* = 3) experiments and 25 larvae per group for each time point. Not significant (ns), *p* > 0.05; * *p* < 0.05; ** *p* < 0.01; *** *p* < 0.001; **** *p* < 0.0001.

**Figure 3 viruses-16-00155-f003:**
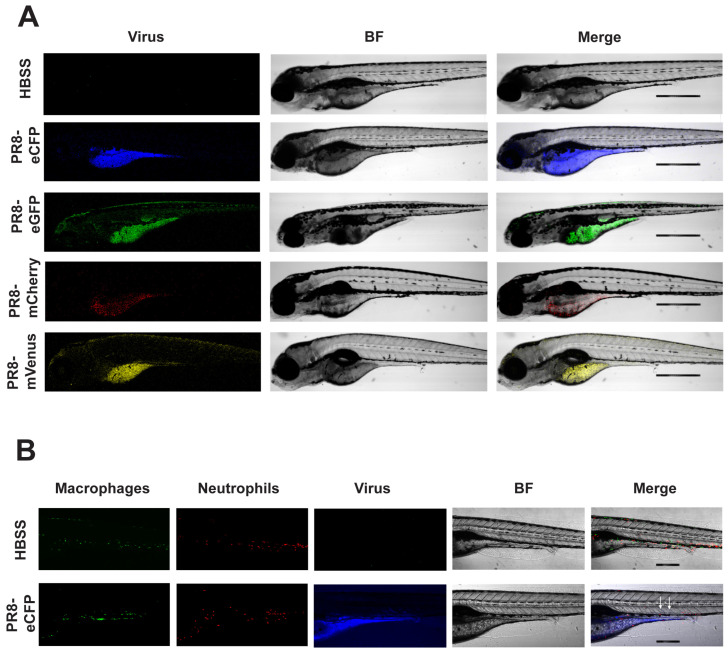
Confocal imaging of Color-flu-infected zebrafish. (**A**) Representative images of larvae at 24 h post systemic injection of HBSS, eCFP-PR8, eGFP-PR8, mCherry-PR8, and mVenus-PR8 at 3 dpf, at 4× resolution. Scale bar: 300 mm. BF: brightfield. (**B**) Representative images of Tg(*mpeg1*:eGFP;*lyz*:dsRed) larvae at 24 h post systemic injection of HBSS, eCFP-PR8 at 3 dpf at 10× resolution, showing macrophages (green), neutrophils (red), and eCFP-PR8 (blue). Infected skeletal muscle was detected (white arrowheads). Scale bar: 700 μm.

**Figure 4 viruses-16-00155-f004:**
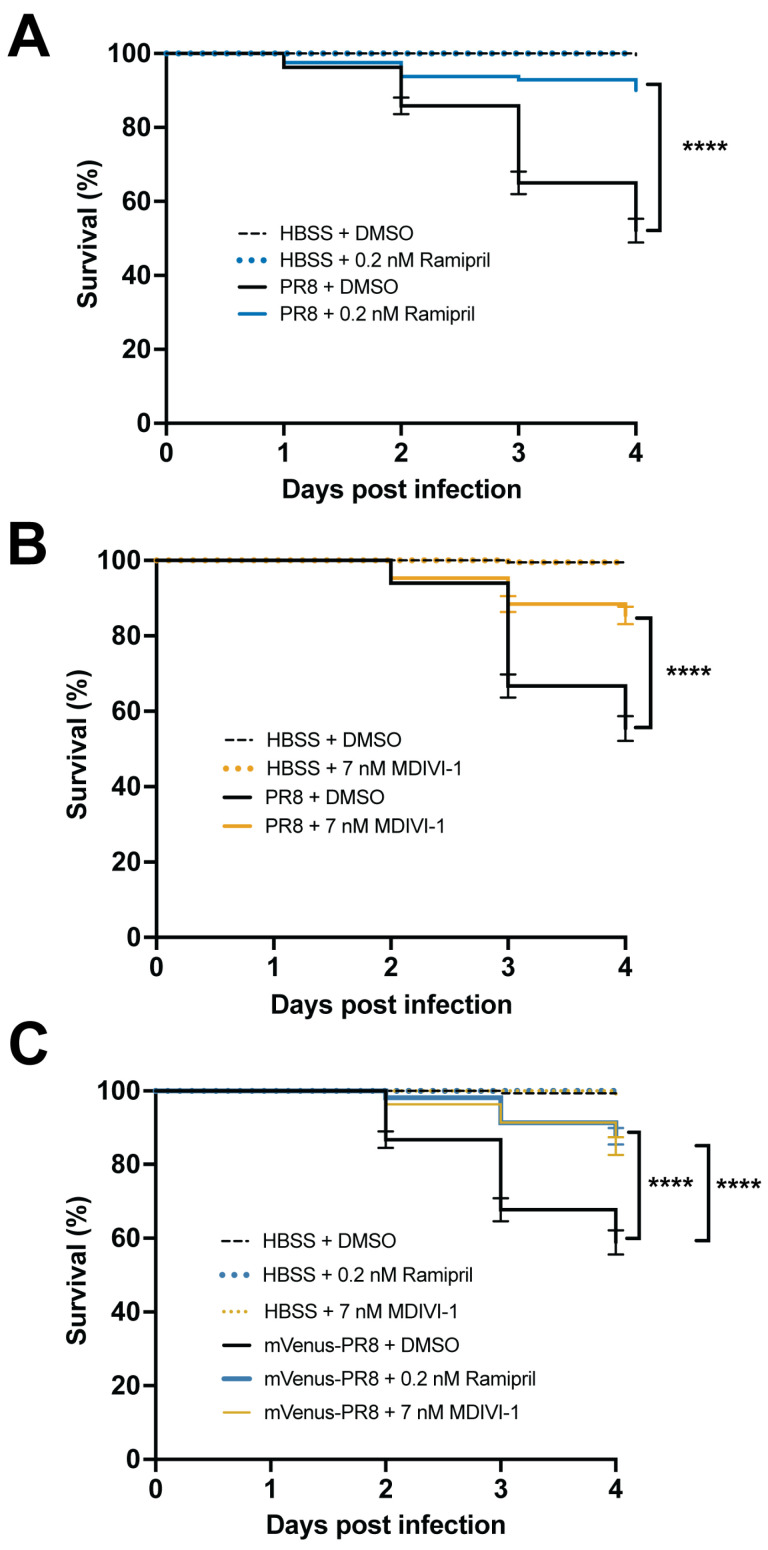
Ramipril and MDIVI-1 increase survival in systemically infected PR8 and Color-flu-infected 3 dpf zebrafish. (**A**) Increased survival in PR8-infected AB zebrafish treated with 0.2 nM ramipril compared to DMSO controls (**** *p* < 0.0001). (**B**) Increased survival in PR8-infected AB zebrafish treated with 7 nM MDIVI-1 compared to DMSO controls (*p* < 0.0001). (**C**) Increased survival in mVenus-PR8-infected AB zebrafish treated with 0.2 nM ramipril or 7 nM MDIVI-1 compared to DMSO controls (*p* < 0.0001). Survival studies were conducted with *n* = 4 and 50 larvae per sample group.

**Figure 5 viruses-16-00155-f005:**
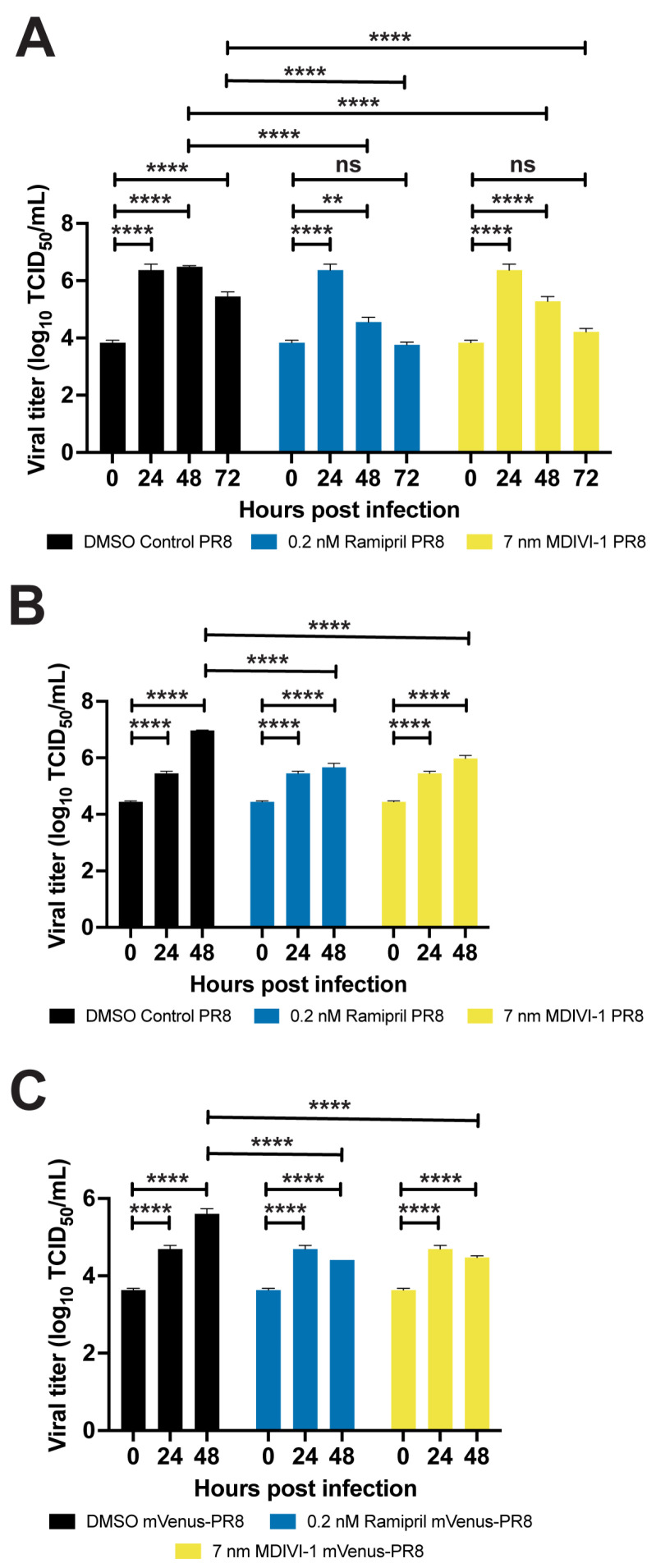
Ramipril and MDIVI-1 treatments lower viral burden in IAV-infected zebrafish. (**A**) Viral titers for both ramipril (0.2 nM)- and MDIVI-1 (7 nM)-treated 2 dpf zebrafish infected with PR8 were significantly higher at 24 and 48 hpi when compared to 0 hpi, but not different at 72 hpi. For the DMSO-treated zebrafish, viral titers were significantly increased at 24, 48, and 72 hpi (*p* < 0.0001 for each comparison). For the ramipril-treated zebrafish, viral titers were significantly increased at 24 (*p* < 0.0001) and 48 hpi (*p* = 0.0022). For the MDIVI-1-treated zebrafish, viral titers were also significantly increased at 24 and 48 hpi (*p* < 0.0001 for each comparison). (**B**) Viral titers for ramipril- and MDIVI-1-treated 3 dpf zebrafish infected with PR8 were significantly lower by 48 hpi (*p* < 0.0001 for each comparison). (**C**) Viral titers for ramipril- and MDIVI-1-treated 3 dpf zebrafish infected with mVenus-PR8 were significantly lower at 48 hpi (*p* < 0.0001 for each comparison). TCID_50_ assays were conducted using *n* = 3 and 25 larvae per group for each time point. Not significant (ns), *p* > 0.05; ** *p* < 0.01; **** *p* < 0.0001.

**Figure 6 viruses-16-00155-f006:**
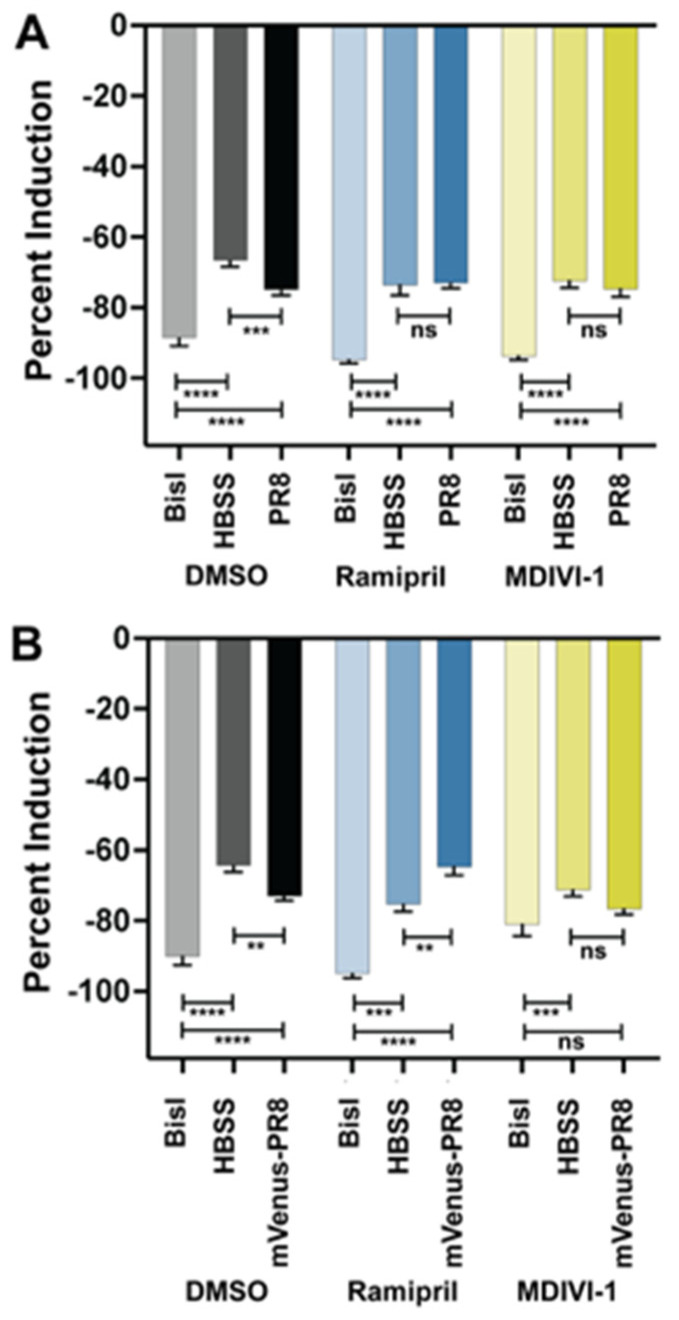
Ramipril and MDIVI-1 treatment alters the respiratory burst response in IAV systemically infected zebrafish at 48 hpi. (**A**) Respiratory burst response in larvae treated with DMSO, ramipril (0.2 nM), and MDIVI-1 (7 nM) at 48 hpi following systemic injection at 3 dpf with PR8 or HBSS. PR8 infection decreased the response over HBSS in DMSO-treated controls (adjusted *p* = 0.0008). Both ramipril and MDIVI-1 treatment remedied the reduction in respiratory burst response, as the PR8-infected larvae had the same response as HBSS-injected controls (comparisons were not significant (ns)). The protein kinase C inhibitor bisindolylmaleimide I (BisI) was used as a positive control as it suppresses the respiratory burst response (adj. *p* < 0.0001 for all comparisons). (**B**) Respiratory burst response in larvae treated with DMSO, ramipril (0.2 nM), and MDIVI-1 (7 nM) at 48 hpi following systemic injection at 3 dpf with mVenus-PR8 or HBSS. Similar to the PR8-infected larvae, mVenus-PR8 infection decreased the response over HBSS in DMSO-treated controls (adj. *p* = 0.0040). Ramipril treatment resulted in a higher respiratory burst response with mVenus-PR8 infection than HBSS controls (adj. *p* = 0.0049). MDIVI-1 treatment remedied the reduction in respiratory burst response as the mVenus-PR8-infected larvae had the same response as HBSS injected controls. The BisI controls were different to the DMSO (adj. *p* < 0.0001 for both), ramipril (adj. *p* = 0.0003 for HBSS, and adj. *p* < 0.0001 for mVenus-PR8), and MDIVI-1 (adj. *p* = 0.0006 for HBSS, ns for mVenus-PR8) groups. Not significant (ns), *p* > 0.05; ** *p* < 0.01; *** *p* < 0.001; **** *p* < 0.0001.

**Figure 7 viruses-16-00155-f007:**
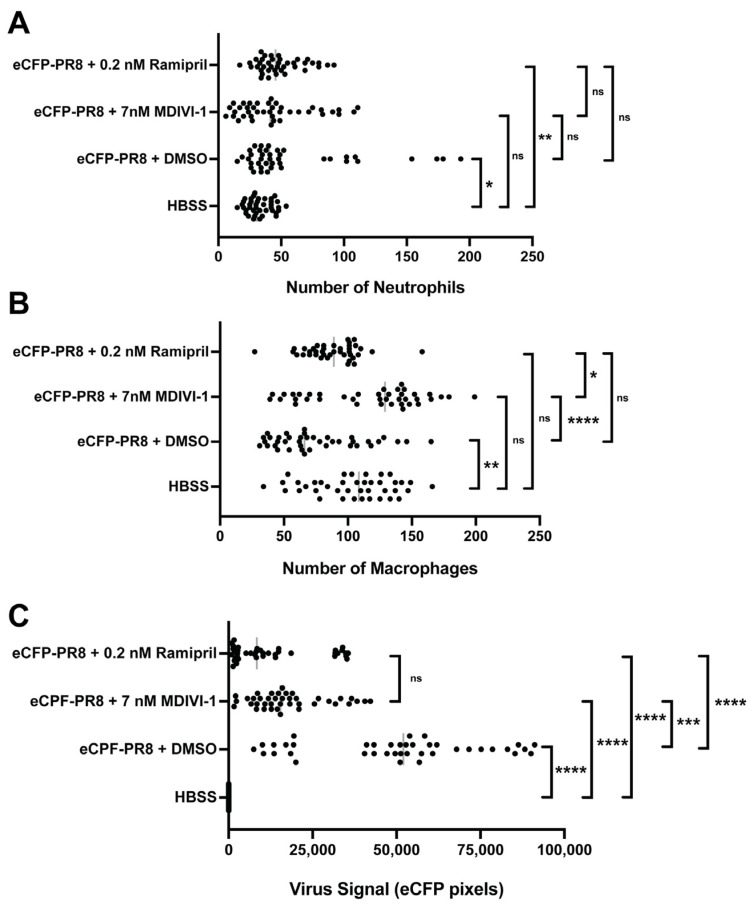
Quantification of the number of neutrophils (**A**), macrophages (**B**), and the relative abundance of virus infected cells (**C**) via fluorescent confocal imaging of Tg(*mpeg1*:eGFP;*lyz*:dsRed) larvae at 48 h post injection by eCFP-PR8 or HBSS following treatment with DMSO, ramipril, and MDIVI-1 per optical cross section. For each larva, 10 five-micron optical cross sections were analyzed, which together totaled 50 microns (*n* = 4 representative larvae). (**A**) The number of neutrophils increased with eCFP-PR8 infection following DMSO treatment over HBSS controls (adjusted *p*-value = 0.0145), and with ramipril treatment (adj. *p*-value = 0.0010), but not with MDIVI-1 treatment. (**B**) The number of macrophages decreased in eCFP-PR8-infected larvae treated with DMSO over HBSS controls (adj. *p*-value = 0.0020), but was not different with ramipril or MDIVI-1 treatment. MDIVI-1 treatment increased the number of macrophages compared to the DMSO controls (adj. *p*-value < 0.0001), and ramipril-treated larvae (adj. *p*-value = 0.0126). (**C**) The extent of viral infection was higher in eCFP-PR8-infected larvae treated with DMSO, ramipril, and MDIVI-1 (adj. *p*-value < 0.0001 for all comparisons). The level of virus infection was lower with ramipril (adj. *p*-value < 0.0001) and MDIVI-1 (adj. *p*-value = 0.0008) treatment compared to the DMSO-treated controls. Not significant (ns), *p* > 0.05; * *p* < 0.05; ** *p* < 0.01; *** *p* < 0.001; **** *p* < 0.0001.

## Data Availability

All relevant data are within the manuscript and its Appendix A.

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
