# Peer review of "Color-Flu Fluorescent Reporter Influenza A Viruses Allow for In Vivo Studies of Innate Immune Function in Zebrafish"

_viruses, 2024, doi:10.3390/v16010155_

Round 1
Reviewer 1 Report
Comments and Suggestions for Authors
In this study, Soos B.L. et al. described Influenza infection model in Zebrafish larvae. Exploiting fluorescent reporter Inluenza A viruses, they investigated the infectious process and the host responses. Reproducing the model established by Gabor K.A. et al. in 2014, they described influenza infection following IV and swimbladder injection at different stages of zebrafish development (2, 3 or 4dpf) in three different genotypes (AB, EK and Casper). Combining confocal imaging, molecular biology and virology, they compared the kinetic of viral replication of four fluorescent IAV strains and exploited the model to assess the antiviral efficacy of ramipril and MDIVI-1.
Major concerns:
The fluorescent reporter Influenza A viruses expressed fluorescent reporter protein fused to the NS1 open reading frame (Fukuyama S. et al., 2015). These reporter systems enable the visualization of influenza virus infected cells in zebrafish larvae. They did not allow the detection of viral particles, which are not fluorescent. This leads to misinterpretation of the results (line 17, 22, 238, 370-371, 452, 455, …) and wrong conclusion of the authors (line 495-498: The zebrafish Color-flu model of IAV infection is unique as it is the only model where interactions between virus particles and host cells in an intact vertebrate can be visualized in vivo using fluorescent confocal imaging. Line 529-530: Confocal imaging of Color-flu infected larvae show virus throughout different tissues of the larvae and Line 592-593 The model is the only IAV model that allows for the visualization of interactions between virus particles and host cells in an intact vertebrate in vivo.
In addition, the micrometric resolution of the confocal microscope used in the study is not sufficient to detect influenza viruses of around 100 nm in size.
The introduction is incomplete and needs to be revised to give a better presentation of the fluorescent IAV strains (genetic constructs, results obtained in mice), the Zf model and its interest in the study of viral pathologies and to include the missing works published on IAV in the Zf model.
The authors do not present multispectral imaging of IAV viruses as mentioned in the title. There must be a confusion as multispectral imaging refers to simultaneously detection of different colors as shown in Fig.2 of Fukuyama S. et al., 2015.
Statistical analyses and/or legends of Figures must be complete to precise the number of independent experiments and to clarify the number of fish per condition (especially for the image analyses: Fig7).
Viral doses used in infectiology experiments must be define by the viral title instead of dilution throughout the text (ex : virus strains were diluted in HBSS at 87% virus and 13% HBSS).
In the characterization of PR8 and color-flu systemically infected AB, did the authors administer the same dose? Figure 1D shows significant variability at t0, making it impossible to compare strains (also mentioned in line 295-296).
The graph needs to be presented differently, perhaps by separating the data by strain. This must facilitate the reading of the statistical results.
Characterization of the viral induced immune response (line 298-309) has been included in the 3.2 part Color-flu infection decreases survival and replicates in Zebrafish, although it constitutes an independent part.
The selection of the proinflammatory genes (cxcl8b, irf9, ncf1, ripk1l, socs3b and tnfa), which have been examined is not really clear. IAV infection triggers antiviral response based on the synthesis of IFN-β upon recognition of pathogen-associated molecules by TLR3 or RIG-I innate immune sensors. Hence, the authors should rather investigate the induction of zebrafish IFN and IFN-stimulated genes well characterized in the zebrafish community.
Line 379-382 must be revised. Next, we evaluated how two small molecule drugs, ramipril and MDIVI-1, would alter the response to IAV infection by examining survival, virus burden, immune capacity by respiratory burst assay, and relative immune cell proliferation and virus load using confocal imaging. Confocal imaging described in the study does not enable to conclude on cell proliferation and virus load. Cell proliferation should be assessed with specific markers and virus load must be correlated with fluorescent signals to be able to conclude.
Minor concerns
Origin of Zf transgenic lines should be precised: Tg(mpeg1:eGFP; lys:dsRed) has been obtained by crossing Tg(mpeg1:eGFP) add a reference and Tg (lys:dsRed) add a reference.
Line 253-255 : It was previously shown that zebrafish express 𝝰-2,6-linked sialic acid-containing receptors and PR8 H1N1 and X-31 A/Aichi/68 H3N2 IAV reduced survival of and replicated within zebrafish with disseminated infection. please, correct the sentence
Line 255-256, We modified the original zebrafish IAV infection protocol to achieve approximately 50% mortality by 7 days post fertilization (dpf) by infecting embryos with PR8 IAV at either 2 or 3 dpi. Please, specify the modification of the original protocol
Line 286, kindly provided by Dr. Yoshihiro Kawaoka should be removed and mentioned in the
material and methods.
Figure 3 should be presented differently to improve the visualization of fluorescent signals over the fish: separate the channels to present fluorescence, bright field and merge.
Line 390 - 394. The ramipril and MDIVI-1 concentrations used were selected after evaluating differences in survival due to drug treatment were evaluated for a range of doses for ramipril (0.1, 0.2, 0.3 and 0.4 nM) and MDIVI-1 (3, 5, 7 and 10 nM) in PR8-infected larvae at either 2 or 3 dpf (Figures S3 and S4). Sentence must be revised.
Comments on the Quality of English Language
Minor editing of English language required
Author Response
Following the very helpful comments and suggestions from the reviewers, we have revised our manuscript, Color-flu Fluorescent Reporter Influenza A Viruses Allow for in vivo Studies of Innate Immune Function in Zebrafish, for publication in Viruses. Our initial manuscript was reviewed as needing “Major Revisions” and we have revised it accordingly.
Here is a summary of the comments from reviewer #1 and our responses.
Major concerns:
- Color-flu allows for visualization of infected cells and not virus particles.
We very much appreciate the reviewer’s point that the Color-flu strains allow for visualization of infected cells and not virus particles. The reviewer correctly describes that the fluorescent reporter is fused to the NS1 open reading frame which is diagrammed in Supplementary Figure 5 of the Fukuyama et al paper. Following infection, host cells express the fluorescent reporter thereby allowing for visualization of the infected cells. We have updated the manuscript accordingly. All changes were tracked.
- Resolution of confocal not sufficient.
We thank the reviewer again for their feedback on Major Concern #1. The manuscript was revised to describe how we quantified virus load by imaging infected cells and not the number of virus particles. The resolution of confocal imaging is sufficient to visualize cells infected with Color-flu like what was originally done by Fukuyama et al.
- Improve introduction to include more detail on Color-flu and zebrafish IAV model.
We agree with the reviewer that more detail about the Color-flu and zebrafish IAV model would improve the manuscript. We added more detail to the introduction about Color-flu, and citations to other studies that used the zebrafish IAV model.
- Multi-spectral imaging of viruses not performed as shown in Fig. 2 of Fukuyama et al.
In the original description of Color-flu by Fukuyama et al, mice were co-infected with multiple Color-flu strains to visualize local spread of a virus in bronchial tissue. Here, we infected zebrafish with individual strains separately in order to demonstrate how Color-flu could be used to infect zebrafish fluorescent reporter strains. These reporter strains typically have green (eGFP) or red (mCherry) reporters which would not be discernible from cells infected with the eGFP-PR8 and mCherry-PR8 Color-flu strains. In future experiments, we plan to perform co-infections in AB zebrafish larvae to determine whether we can also visualize local viral spread. We modified the title of the manuscript by substituting “Color-flu” for “multi-spectral” to address the reviewer’s comment.
- Clearly state the number of independent experiments and sample sizes, especially for Fig. 7.
We have revised the manuscript to include information about the samples size for Figure 7 as well as other experiments in the figure legends and methods sections as appropriate.
- Define viral doses used as virus titers instead of dilution.
We agree with the reviewer that the viral doses need to be described and we added detail about the viral doses used to the methods.
- Clarify level of infection used for Figure 1D. Concern about variability at time 0 making it “impossible to compare strains”.
We reviewed our data and the original calculations used in the analysis was done with the wrong values for the serial dilution at time 0. We corrected these calculations and revised Figure 1D accordingly.
- Figure 1D needs to be presented differently so that it is separated by strain in order to facilitate interpretation of the statistical results.
We have reorganized Figure 1D as the reviewer suggested. We also made similar changes to Figure 2C, 2D, and 2F, as well as Figure 5A, 5B and 5C.
- Move gene expression studies from section 3.2 to a new section.
As suggested by the reviewer, we created a new section to describe the gene expression studies (see the new Section 3.3).
- Clarify selection of proinflammatory genes. Add interferon and interferon-stimulated genes.
The reviewer is correct that interferon is induced upon infection in zebrafish larvae. We added more detail about the selection of the genes as suggested by the reviewer.
- Revise lines 379-382 to not talk about cell proliferation or virus load. Cell proliferation needs to be measured using markers, and virus load needs to be correlated with fluorescent signals.
We agree with the reviewer that we measured differences in cell number and not cell proliferation. We have revised the sentence accordingly (see first sentence of section 3.6).
Minor concerns:
- Add references for Tg(mpeg1:eGFP) and Tg(lyz:dsRed).
As suggested by the reviewer, we added citations for the Tg(mpeg1:eGFP) and Tg(lyz:dsRed) fluorescent reporter lines in the Materials and Methods section.
Tg(mpeg1:eGFP)
Renshaw SA, Loynes CA, Trushell DM, Elworthy S, Ingham PW, Whyte MK. A transgenic zebrafish model of neutrophilic inflammation. Blood. 2006 Dec 15;108(13):3976-8. doi: 10.1182/blood-2006-05-024075. Epub 2006 Aug 22. PMID: 16926288.
Tg(lyz:dsRed)
Hall C, Flores MV, Storm T, Crosier K, Crosier P. The zebrafish lysozyme C promoter drives myeloid-specific expression in transgenic fish. BMC Dev Biol. 2007 May 4;7:42. doi: 10.1186/1471-213X-7-42. PMID: 17477879; PMCID: PMC1877083.
- Reword lines 253-255.
We revised this sentence as suggested by the reviewer.
- Describe original protocol near lines 255-256.
As suggested by the reviewer, we provided more detail about the original protocol.
- Move acknowledgement of Dr. Kawaoka on line 286 to materials and methods.
As suggested by the reviewer, we removed the acknowledgement from the Results section.
- Improve Figure 3 to show fluorescence, bright field, and then merge.
We revised the figure as suggested by the reviewer.
- Revise lines 390-394.
We revised this sentence as suggested by the reviewer.
Reviewer 2 Report
Comments and Suggestions for Authors
Authors of presented study has developed an important alternative to mammalian models for the study of host pathogen interactions. The Color-flu zebrafish model of influenza virus infection can be used to study the innate immune response to IAV. Authors studied the role of macrophages and neutrophils during IAV infection. Using Color-flu, they characterized how ACE inhibition by ramipril and mitophagy inhibition by MDIVI-1 both improve the response to IAV infection by limiting inflammation through distinct mechanisms. The manuscript is well written, the obtained data are clearly presented. The manuscript can be accepted in the present form.
Author Response
We appreciate the reviewer's comments on our manuscript. As this reviewer thought the manuscript could be accepted without revisions, we do not have any specific comments from them to address.
Reviewer 3 Report
Comments and Suggestions for Authors
This is very valuable work which allows for the visualization of interactions between virus particles and host cells in an intact vertebrate in vivo. This opens up completely ne possibilities to observe,track an counter the virus attack. The work is done very carefully. Materials and methods are described very precisely, which allows them to be reproduced in another laboratory. The results are thoroughly discussed.
The manuscript is masterfully written based on well-chosen literature.
Author Response
We appreciate the reviewer's comments on our manuscript. The reviewer did not have any specific comments to address.
Round 2
Reviewer 1 Report
Comments and Suggestions for Authors
The edited manuscript has been improved, although it still needs major revision.
The study describes Zf model for in vivo study of color flu infection and characterization of the viral induced innate immune responses. However it does not provide results on interaction between innate immune and infected cells as the authors point out.
The manuscript must be revised accordingly (lines 22-23, 547, 565, 641-642)
Major concenrs
lines 333-334 : "Significant increases in viral titers were observed at 24 hpi for all Color-flu strains except eGFP-PR8" does not correspond to Figure 1D, which shows the increase of viral titers for eGFP-PR8.
lines 336-352: the study of proinflammatory gene expression should be improved adding type I IFN and ISGs, major components of the antiviral response.
lines 363-366: the pourcentages of survival presented in the text do not correspond to the results presented in Figure 2. This must be corrected.
Figure 2D. The statistical analysis must be revised (differences > 1log between 0 and 24hpi).
Figure 3 (A and B). Readability of the fluorescence channels should be improved 1/ adjusting brightness and contrast and 2/adding arrowheads to show the fluorescence signals "observed in the head and skeletal muscle" (line 409).
The legend should precise that the confocal images correspond to representative larvae (lines 418-422).
Figure 3B. The figure must be completed adding the eCFP chanel of the HBSS condition.
Figure 4. if the untreated (DMSO) individuals presented in panels (A and B) and (C and D) are the same, then the panels must be combined.
Figure 7. Image analyses performed in Figure 7 must be precised. What does each point on the graph correspond to? the number of macrophages/neutrophils or pixels per single individual? per optical section? This needs to be clarified.
Minor concerns
lines 168-169; lines 325-326 :the infectious dose should be indicated in terms of viral titer rather than volume
lines 319-320; lines 401-402: n=3. Does that mean that the authors performed 3 independant experiments? This needs to be clarified.
lines 563-565; "The combination of sequence variants found in casper could potentially explain the difference in response to IAV infection." should be better discussed to present the arguments in absence of data supporting this hypothesis in the litterature.
Author Response
Dear Editors of Viruses:
Following very helpful comments and suggestions from the reviewer #1 from our first resubmission, we have revised our manuscript, Color-flu Fluorescent Reporter Influenza A Viruses Allow for in vivo Studies of Innate Immune Function in Zebrafish, for publication in Viruses.
Here is a summary of the comments from reviewer #1 and our responses.
Major concerns:
- The zebrafish model of influenza A virus (IAV) infection does not provide results on the interaction between innate immune and infected cells.
We very much appreciate the reviewer’s point. We revised the manuscript accordingly to emphasize that the model allows for simultaneous visualization of IAV-infected cells and host immune cells.
- Lines 333-334: "Significant increases in viral titers were observed at 24 hpi for all Color-flu strains except eGFP-PR8" does not correspond to Figure 1D, which shows the increase of viral titers for eGFP-PR8.
In our last revision we had updated the figure legend for Figure 1D but did not make the corresponding change in the text. We revised the last sentence of Section 3.2 appropriately.
- Lines 336-352: The study of proinflammatory gene expression should be improved adding type I IFN and ISGs, major components of the antiviral response.
Section 3.3 and Supplemental Figure S2 examines the expression of six genes. One of those genes, irf9, was studied because it is an upstream regulator of interferon signaling during viral infection. We respect the opinion of the reviewer that there are many possible genes that could be assayed. However, it is beyond the scope of this manuscript to broadly examine gene expression patterns following IAV infection.
- Lines 363-366: the percentages of survival presented in the text do not correspond to the results presented in Figure 2.
The survival percentages that were listed for Figure 2A and 2B were reversed in the text and we corrected the values.
- In Figure 2D, the statistical analysis must be revised (differences > 1log between 0 and 24hpi).
In Figure 2D the reviewer is correct that the differences between groups were larger than 1 log10 unit. However, five comparisons had adjusted p-values from Dunnett’s multiple comparisons test that were not all significant as indicated in Figure 2D. Two were marginally significant, but not less than 0.05. Here is the list of adjusted p-values for the comparisons in Figure 2D.
-
-
- AB 0 vs. 24 hpi adj. p-value = 0.2562 (not significantly different)
- AB 0 vs. 48 hpi adj. p-value = 0.0524 (not significantly different)
- AB 0 vs. 72 hpi adj. p-value = 0.0323 (significantly different)
- EK 0 vs. 24 hpi adj. p-value = 0.1605 (not significantly different)
- EK 0 vs. 48 hpi adj. p-value = 0.0525 (not significantly different)
- EK 0 vs. 72 hpi adj. p-value = 0.0211 (significantly different)
- casper 0 vs. 24 hpi adj. p-value = 0.5848 (not significantly different)
- casper 0 vs. 48 hpi adj. p-value = 0.0063 (significantly different)
- casper 0 vs. 72 hpi adj. p-value = 0.0452 (significantly different)
-
- In Figure 3 (A and B), the readability of the fluorescence channels should be improved by: 1) adjusting brightness and contrast and 2) adding arrowheads to show the fluorescence signals "observed in the head and skeletal muscle" (line 409).
We revised Figure 3 as the reviewer suggested. These changes including adding arrowheads to point out infected skeletal muscle in Figure 3B. The text was also changed to state that infection was observed in skeletal muscle as shown in Figure 3B.
- The legend for Figure 3 should precisely state that the confocal images correspond to representative larvae (lines 418-422).
We revised the legend for Figure 3B to state that these were representative larvae.
- Figure 3B must be completed adding the eCFP channel of the HBSS condition.
We revised Figure 3 as the reviewer suggested.
- In Figure 4, if the untreated (DMSO) individuals presented in panels A and B, and C and D are the same, then the panels must be combined.
The controls for Figure 4 panels A and B were separate, but the controls for panels C and D were the same. Following the reviewer's suggestion, we combined the data for panels C and D into a new panel C and revised the figure legend appropriately.
- In the description of Figure 7, the image analyses performed must be precisely described. What does each point on the graph correspond to? the number of macrophages/neutrophils or pixels per single individual? Per optical section? This needs to be clarified.
Following the reviewer's suggestion, we revised the text and legend for Figure 7 to clarify the analysis done on the analyzing optical cross sections. Each point in Figure 7 represents the quantification of one cross section. For each larvae, 10 five micron optical cross sections were analyzed that together totaled 50 microns (n = 4 representative larvae).
Minor concerns:
- The infectious dose should be indicated in terms of viral titer rather than volume (lines 168-169; lines 325-326).
We added the viral titers for the four Color-flu strains that we measured using TCID50 assays to section 2.4 (methods), and results (section 3.2).
- For lines 319-320 and 401-402, does n = 3 mean that the authors performed 3 independent experiments?
Yes, these were independent experiments, and we revised the text in sections 3.1 to clarify this point.
- For lines 563-565; "The combination of sequence variants found in casper could potentially explain the difference in response to IAV infection." should be better discussed to present the arguments in absence of data supporting this hypothesis in the literature.
As suggested by the reviewer, we added more detail about the sequence variants and gene expression differences found in casper mutants by Bian et al [reference #45] in the second paragraph of the discussion. The study by Bian et al conducted whole genome sequencing to find sequence variants and RNA sequencing studies of skin and skeletal muscle in adult AB (wild-type), casper and roy mutants. We examined the results of the RNA sequencing studies provided by Bian et al in their supplemental tables S7 and S8 and found that sting1, an activator of interferon signaling, was consistently upregulated in casper mutants compared to AB controls. We added two sentences to describe these differences found in casper mutants.
Thank you for considering and reviewing our revised manuscript for publication in Viruses.